# Determinants of overweight and obesity among children between 5 to 11 years in Ecuador: A secondary analysis from the National Health Survey 2018

Betzabé Tello [1◉], José Ocaña[2◉], Paúl García-Zambrano[3‡], Betsabé Enríque-Moreira[3‡], Iván Dueñas-Espín[2◉]*

1 Center for Research in Health in Latin America CISeAL, Pontificia Universidad Católica del Ecuador, Quito, Ecuador, 2 Instituto de Salud Pública, Facultad de Medicina, Pontificia Universidad Católica del Ecuador, Quito, Ecuador, 3 Postgrado de Medicina Familiar, Facultad de Medicina, Pontificia Universidad Católica del Ecuador, Quito, Ecuador

◉ These authors contributed equally to this work.
‡ PGZ and BEM also contributed equally to this work.
* igduenase@puce.edu.ec

**Data Availability Statement:** All relevant data are within the paper and its Supporting Information files.

## Abstract

### Background

During the 1990s, global eating habits changed, affecting poorer and middle-income nations, as well as richer countries. This shift, known as the "obesity transition," led to more people becoming overweight or obese worldwide. In Ecuador, this change is happening, and now, one in three children is affected by overweight or obesity (OW/OB). This study explores the links between social, economic, and demographic factors and childhood obesity in Ecuador, seeking to provide insights for shaping future health policies in response to this intricate shift.

### Methods

A cross-sectional study using 2018 National Health and Nutrition Survey data from Ecuador. Weighted percentages were computed, and odds ratios for OW/OB unadjusted and adjusted for each category of explanatory variables were estimated using multilevel multivariate logistic regression models.

### Results

Among 10,807 Ecuadorian school children aged 5 to 11, the prevalence of OW/OB was 36.0%. Males exhibited 1.26 times higher odds than females (95% CI: 1.20 to 1.33), and each additional year of age increased the odds by 1.10 times (95% CI: 1.09 to 1.10). Economic quintiles indicated increased odds (1.17 to 1.39) from the 2nd to 5th quintile (the richest) compared with the first quintile (the poorest). Larger household size slightly reduced odds of OW/OB (adjusted odds ratio [aOR] = 0.93, 95% CI: 0.91 to 0.95), while regular physical activity decreased odds ([aOR] = 0.79, 95% CI: 0.75 to 0.82). The consumption of

**Funding:** The author(s) received no specific funding for this work.

**Competing interests:** The authors have declared that no competing interests exist.

school-provided meals showed a non-significant reduction (aOR: 0.93, 95% CI: 0.82 to 1.06). Children from families recognizing and using processed food labels had a higher likelihood of being overweight or obese (aOR = 1.14, 95% CI: 1.02 to 1.26).

## Conclusion

Age, male gender, and higher economic quintile increase OW/OB in Ecuadorian school children. Larger households and physical activity slightly decrease risks. Ecuador needs policies for healthy schools and homes, focusing on health, protection, and good eating habits.

## Introduction

Similar to high-income countries, low- and middle-income countries-initiated processes of nutritional and food environment transition in the 1990s [1]. These transitions are characterized by an increase in the consumption of processed foods, edible oils, and sugary beverages, as well as a greater tendency to eat outside the home and an increased availability of ultra-processed products. All these changes have come at the expense of healthy and traditional diets. Simultaneously, the population in these countries has gradually reduced physical activity and increased sedentary behavior [1–4].

These transitions have led to a gradual increase in overweight and obesity (OW/OB) in all age groups, with a special increase in childhood OW/OB [2–5]. This global increase follows a pattern known as the "obesity transition" [6]. This pattern is characterized by a gradual shift in the burden of OW/OB from high-income to low- and middle-income countries, from wealthy households to poor ones, from urban to rural areas, and from adults to children. This changes affect several countries in Latin America [6].

In Ecuador, the prevalence of childhood OW/OB surged by almost 5 percentage points from 2012 to 2018, reaching 35.4% [7]. This alarming increase predominantly impacts urban areas, males, those with mixed and white ethnic backgrounds, and wealthier households. Importantly, this trend extends beyond, affecting middle- and low-income households and rural populations, highlighting the urgency for a thorough exploration of its determinants [7].

While there is extensive knowledge about the social, environmental, and clinical determinants of excess malnutrition in school-age children, particularly in high-income countries [8], few countries in the region have studied these determinants [9, 10]. In low- and middle-income countries such as Ecuador, there is a lack of national information regarding the determinants of childhood obesity in school-age children, except for some localized studies [11–13]. The existing knowledge gap precludes the development of effective public health policies. Despite the bulk of scientific evidence, the Ecuadorian government's efforts are not focused on improving school environments, increasing taxes on sugary beverages, or improving the labelling of processed and ultra-processed products; rather, it has reduced taxes on ultra-processed food [14]. Therefore, initiatives with proper implementation, supervision, and robust evaluations are necessary to demonstrate their impact and cost-effectiveness in school-age populations. These insights will serve as a compass for evidence-based public policies and interventions, which are crucial for combating childhood obesity in Ecuador.

Childhood obesity is a precursor to adult cardiovascular diseases and cancer [15–17]. Understanding its determinants is vital for making informed public health policy decisions. This study aims to identify the independent factors associated with obesity and overweight in

Ecuadorian school-age children (5–11 years). By delving into obesogenic environments and contextual sociodemographic conditions, this research offers valuable territorial insights.

## Materials and methods

### Study design

This cross-sectional study involved a secondary analysis, utilizing data from the 2018 National Health and Nutrition Survey of Ecuador. To ensure methodological rigor and transparency in both the study design and dissemination of findings, compliance with the Strengthening the Reporting of Observational Studies in Epidemiology (STROBE) [18] guidelines was scrupulously maintained, as detailed in S1 Table.

### Population and sample

In our study, we included data from children encompassing sociodemographic information, anthropometric measurements, dietary habits at home and school, and physical activity status. Additionally, we integrated information about whether the children's households identified, understood, and used the nutritional traffic light labelling system for processed foods and beverages that was implemented in Ecuador. These questions were incorporated into the National Health and Nutrition Survey of 2018 (see flowchart in Fig 1). We included study subjects who: (i) are ≥5 years old or ≤11 years old; (ii) had complete anthropometric information, and (iii) had complete data regarding age, ethnicity, economic quintile, schooling characteristics, dietary habits, and physical activity.

### The ENSANUT 2018 survey

The Ecuadorian National Health and Nutrition Survey 2018 (ENSANUT 2018, for its acronym in Spanish) was a cross-sectional study conducted in 2018 that involved nationally representative samples from the Ecuadorian population [19].

In the ENSANUT 2018 study, a two-stage sampling strategy was employed to secure a representative sample of the Ecuadorian populace. Initially, Primary Sampling Units (PSU) were chosen through stratified sampling, incorporating proportional probability to size. Subsequently, an average of 18 households per PSU were randomly selected for investigation. Within these households, specific demographic groups were identified. For households with children aged 5 to 11 years, a qualified child informant was selected for interview and asked to complete a specialized questionnaire. The abovementioned sampling approach ensured the data quality and representativeness of the study. Further information on the methodology, datasets, and findings of ENSANUT 2018 is available at: https://www.ecuadorencifras.gob.ec/institucional/home/.

### Measurements

We used the information that the survey collected about sex, age of the child, ethnicity, education of the children, economic quintile, regular class attendance, geographical regions of Ecuador, receiving the human developing bonus (BDH, for its acronym in Spanish), number of people in the household, disposal of excreta, physical activity, perception of consumption of vegetables, consumption of fast food, days per week of school food consumption, buying food at school, consumption of the food provided by the school, recognizing, understanding, and using the nutritional traffic light labelling of processed foods; and, consumption of processed foods with a red label. The Ecuadorian Nutritional Traffic Light Labelling system uses three colours—red, yellow, and green—to indicate the levels of sugar, fat, and salt in processed

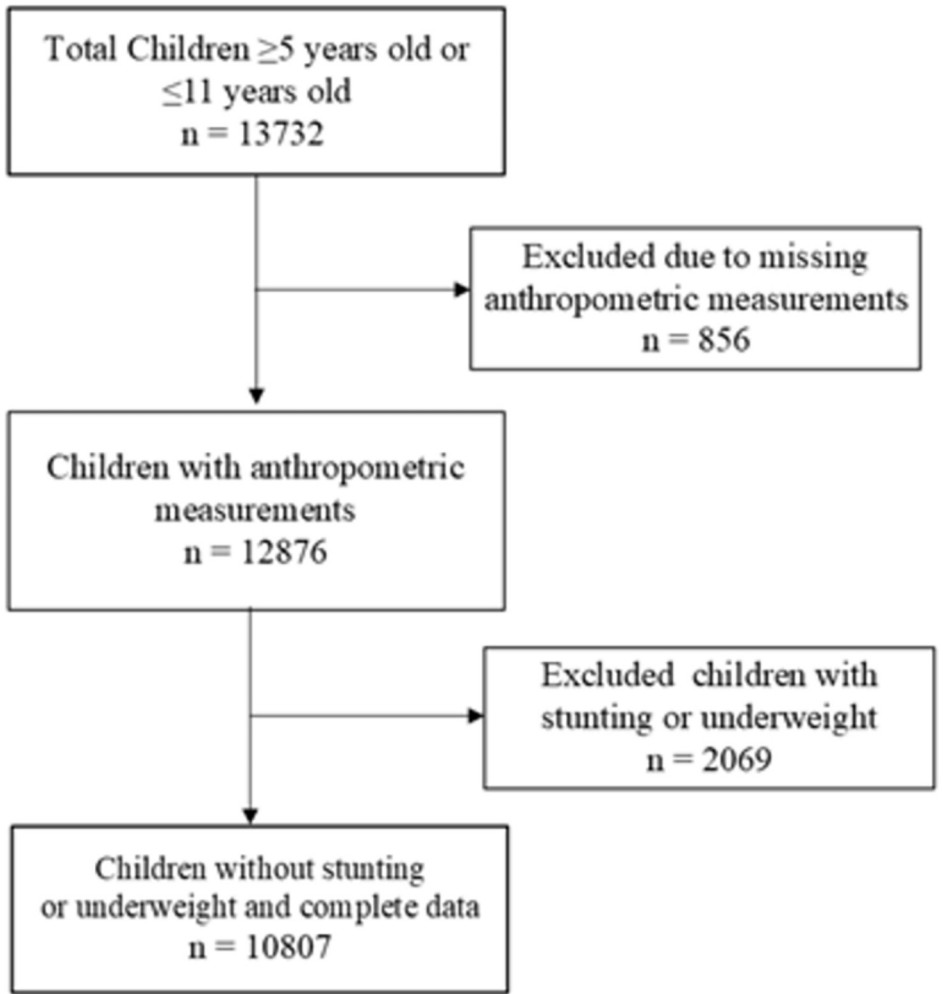

**Fig 1. Flowchart showing the study population and selection of study participants.**

foods. Red signifies high concentrations, yellow denotes medium levels, and green indicates low content [20]. This system empowers consumers to make healthier food choices.

## Main outcome

The main outcome variable was, overweight or obesity. The World Health Organization (WHO) macro program Stata (WHO AnthroPlus) was used to establish the nutritional status of children based on WHO 2007 standards for the classification of children as overweight or obese between 5 and 19 years. Overweight was defined as a Body-Mass-Index (BMI)-for-age greater than 1 standard deviation above the WHO Growth Reference median; and obesity as a greater than 2 standard deviations above the WHO Growth Reference median [21].

## Statistical analyses and sample considerations

A priori, we calculated that a sample of 9759 individuals is enough to estimate, with 95% confidence and an accuracy of +/- 1 percentage units, a population percentage that will predictably be around 35.38% [7]. The percentage of necessary replacements is expected to be 10%.

In the analysis conducted, the 'svy' function of Stata 16.1 (StataCorp. 2019. Stata Statistical Software: Release 16. College Station, TX: StataCorp LLC.) was employed, which is adept at tailoring calculations to the specific structure of our survey data. Adjusted percentages for categorical data and means with standard errors for continuous data are computed by this function, taking into account the sample design. Furthermore, it has been ensured that the 'svy' function facilitates the fitting of complex statistical models, conforming the results to the survey settings predefined by svyset. Following this, we compared the characteristics of children who are not overweight or obese with those who are, ensuring a methodical approach for a balanced and representative analysis. To assess the differences between groups, we employed Pearson's chi-squared test for categorical variables and the z test for numerical variables.

Then, multilevel logistic regression models were employed to discern the associations between independent variables and the prevalence of overweight and obesity (OW/OB). These models facilitated the estimation of both unadjusted and adjusted odds ratios (OR and aOR, respectively), confidence intervals (95%CI) of ORs, and their corresponding p-values for each independent variable, offering insights into the likelihood of OW/OB presence relative to each variable or their respective categories. To enhance the precision of our analysis, we incorporated geographical regions as a level within our models. This was imperative due to the observed variance in OW/OB prevalence among different regions. By doing so, we could account for regional disparities that may influence the health outcome. Additionally, we integrated expansion factors in our estimations to ensure congruence with the stratified sampling design and the primary sampling units. This stratification was an essential step in managing the inherent variability and correlations within our sampled groups, thereby ensuring that our estimates remained robust and representative of the broader population.

We started by creating a saturated model that considered all possible variables. Next, we removed any variable that did not show a strong enough link to the outcomes we were interested in; specifically, with a p-value of 0.05 or higher [22], indicating a less than 5% chance that the factor was meaningfully related to the outcome. After trimming down the model in this way, we were left with a parsimonious model that only included the most relevant variables. Finally, we compared the saturated model with parsimonious one and selected the best model based on the likelihood ratio test that measure how well each model predicts our outcomes of interest. The final model was stratified by sex. Given the small number of missing data (there were missing values in <1% of the whole database), we employed complete case analysis to estimate statistical associations.

To test for potential effect modification, we performed several secondary analyses to assess the sensitivity of our estimates with our assumptions regarding biases, and to test for model misspecifications. We ran the final model excluding: (i) children categorized in the highest income quintile, (ii) children whose parents received the BDH, and (iii) children within the upper third of the highest number of people per household.

### Ethical issues

The research protocol was thoroughly reviewed and approved by the Ethics Subcommittee for Research in Human Beings of the Faculty of Medicine of the Pontifical Catholic University of Ecuador under the code SB-CEISH-POS-691. The committee determined that informed consent was not required for this study.

### Results

A comprehensive demographic and health snapshot of children aged 5 to 11 years is presented in Table 1. In the study sample comprising 10807 children aged 5 to 11 from Ecuador, it was

**Table 1. Description of the sample.**

| Variable | Whole sample (n = 10807) | Weighted percentage or mean (SE) |
|---|---|---|
|  | No. |  |
| Sex (Male) | 5541 | 50.3 |
| Age of the children | 10807 | 8.0 (0.03) |
| Ethnicity |  |  |
| Ethnicity (Indigenous) | 1252 | 7.2 |
| Ethnicity (Afroecuadorian) | 459 | 4.8 |
| Ethnicity (Mestizo-mixed ethnic background) | 8536 | 80.8 |
| Ethnicity (White) | 141 | 1.3 |
| Ethnicity (Montubio or other) | 419 | 5.9 |
| Economic quintiles by income[a] |  |  |
| Economic quintile by income (1st quintile) | 2865 | 25.4 |
| Economic quintile by income (2nd quintile) | 2344 | 23.6 |
| Economic quintile by income (3rd quintile) | 2096 | 20.4 |
| Economic quintile by income (4th quintile) | 1767 | 16.7 |
| Economic quintile by income (5th quintile) | 1608 | 13.9 |
| Education of children (Elementary school ongoing) | 10618 | 98.3 |
| Regular class attendance (Yes) | 10764 | 99.6 |
| The head of the household receives the BDH (Yes) | 455 | 4.4 |
| Number of people in the household | 10807 | 5.0 (0.03) |
| Inadequate disposal of excreta (Yes) | 2570 | 24.1 |
| Regular physical activity (Yes) | 1389 | 13.5 |
| Perception of low consumption of vegetables (Yes) | 5218 | 47.1 |
| Days per week of consumption in fast food restaurants | 10807 | 0.9 (0.02) |
| Days per week of school food consumption | 7930 | 4.4 (0.03) |
| The child buys food at school (Yes) | 7271 | 64.1 |
| The child eats the food provided by the school (Yes) | 7103 | 64.3 |
| Family members recognize, understand, and use the labelling of processed foods (Yes) | 6635 | 65.4 |
| In the family, they reduced the consumption of processed foods with a red label (Yes) | 4256 | 48.4 |
| Overweight or obesity[b] | 3931 | 36.0 |

BDH = Human Development Voucher, by its Spanish spelling.

SE = Standard error.

[a] Income quintiles are calculated at the household level using monetary labour income per capita, first calculating the total income for each income earner. This total income includes earnings from work, income from investments, transfers, and other benefits, such as cash social transfers. Once we add all these up, we obtain the total household income. Then, we determine the average income per person (per capita income) by dividing the total household income by the number of people in each household. Subsequently, the population is systematically arranged on the basis of the per capita income variable. The calculation of the quintiles was performed by dividing the population into five equal groups, known as quintiles. The first quintile includes the percentage of households with the lowest income, the second quintile includes the next percentage, and so on until the fifth quintile, which includes the percentage of households with the highest income.

[b] Overweight and obesity were determined by calculating the Body Mass Index (BMI), adjusted for age and sex, according to the WHO growth references. Overweight is BMI-for-age greater than 1 standard deviation above the WHO Growth Reference median; and obesity is greater than 2 standard deviations above the WHO Growth Reference median.

found that the mean age of the children was 8.0 years (standard error [SE]: 0.03). About ethnicity, Mestizo (mixed ethnic background) children were predominant, making up 80.8% of the sample. When stratified by economic quintiles, approximately a quarter of the children (25.4%) belonged to the lowest income quintile. Educationally, it was noted that the vast majority (98.3%) were currently attending elementary school. Regarding dietary habits, a significant proportion of the sample (64.3%) reported consuming food provided by the school, and nearly half (48.4%) indicated a reduction in the consumption of processed foods with a red label. Notably, 36.0% of the children were identified as being overweight or obese, according to the Body Mass Index (BMI) adjusted for age and sex, using the WHO Growth References.

We found distinct differences between non-overweight or non-obese children and their overweight or obese counterparts (S2 Table). In our bivariate crude comparison of characteristics between non-overweight/non-obese children (n = 6876) and those identified as overweight or obese (n = 3931), we observed several notable differences. Among these, the average age of children classified as overweight or obese was marginally higher, at 8.3 years, compared to 7.9 years for their non-overweight peers. This age discrepancy was statistically significant, as evidenced by a p-value of less than 0.001 in the z test. Ethnic distribution showed that 81.7% of the overweight or obese groups were Mestizo (mixed ethnic background), compared with 80.3% in the non-overweight group (p-value = 0.602, Pearson's chi-squared). Economic stratification revealed that 20.6% of children were from the lowest income quintile among the overweight or obese compared to the 28.2% among those non-overweight (p-value < 0.001, Pearson's chi-squared). Furthermore, a greater proportion of overweight or obese children (65.7%) consumed food provided by the school, in contrast to 63.2% of the non-overweight children (p-value = 0.124, Pearson's chi-squared). Finally, a difference was detected in the perception of reduced consumption of processed foods with a red label, with 50.4% of overweight or obese children indicating consumption, compared to 47.2% of their non-overweight peers (p-value = 0.097, Pearson's chi-squared).

After running multivariate logistic regression models, we found that, several factors were significantly associated with childhood overweight and obesity (Table 2). According to the final adjusted model, several variables exhibited statistically significant associations between several explanatory variables and being overweight or obese. Notably, male children exhibited a higher likelihood of being OW/OB, with 1.26 times increased adjusted odds (95% CI: 1.20 to 1.33) compared to female children. In addition, for every yearly increase in a child's age, the odds of being overweight or obese increased by 1.10 times (95% CI: 1.09 to 1.10). When broken down by ethnicity, compared to Indigenous children, the Afroecuadorian ethnicity presented a slightly elevated but not statistically significant odds of 1.12 (95% CI: 0.99 to 1.26), while Mestizo children showed 1.14 times increased odds (95% CI: 1.04 to 1.25). White children and those from Montubio (mixed ethnic background of coastal Ecuador) or other ethnicities did not demonstrate statistically significant differences in this model. When considering economic quintiles by income, children in the 2nd quintile demonstrated 1.17 times higher odds (95% CI: 1.07 to 1.31), those in the 3rd quintile showed 1.33 times (95% CI: 1.11 to 1.59), in the 4th quintile it was 1.39 times (95% CI: 1.18 to 1.65), and in the 5th quintile, the odds were 1.39 times higher (95% CI: 1.29 to 1.51) compared to those in the 1st quintile. An increase in the number of household members corresponded to a slight reduction in odds by 0.93 times for each additional person (95% CI: 0.91 to 0.95). Moreover, children with inadequate disposal of excreta exhibited 0.82 times lower odds of being overweight or obese (95% CI: 0.76 to 0.90). Similarly, regular physical activity was associated with reduced odds, at 0.79 times (95% CI: 0.75 to 0.82). Interestingly, children from families that recognized and used processed food labels exhibited a higher likelihood of being overweight or obese, with an adjusted odds ratio

**Table 2. Crude and adjusted Odds Ratios of overweight or obesity from each explanatory variable using multilevel and logistic regression models.**

| Variable | Crude models | | Adjusted models | | | |
|---|---|---|---|---|---|---|
| | OR (IC95%) | p-value | Saturated aOR (IC95%) | p-value | Parsimonious aOR (IC95%) | p-value |
| Male sex (female is the ref.) | 1.24 (1.18 to 1.31) | <0.001 | 1.23 (1.08 to 1.40) | 0.002 | 1.26 (1.20 to 1.33) | <0.001 |
| Age of the child (per each increase in one year) | 1.10 (1.08 to 1.12) | <0.001 | 1.09 (1.08 to 1.10) | <0.001 | 1.10 (1.09 to 1.10) | <0.01 |
| Ethnicity | | | | | | |
| Ethnicity (Indigenous is the ref.) | 1 | - | 1 | - | 1 | - |
| Ethnicity (Afroecuadorian) | 1.09 (0.90 to 1.32) | 0.381 | 0.99 (0.86 to 1.15) | 0.917 | 1.12 (0.99 to 1.26) | 0.062 |
| Ethnicity (Mestizo—mixed ethnic background) | 1.25 (1.06 to 1.49) | 0.009 | 1.06 (0.88 to 1.27) | 0.560 | 1.14 (1.04 to 1.25) | 0.004 |
| Ethnicity (White) | 1.39 (1.05 to 1.85) | 0.021 | 1.17 (0.80 to 1.71) | 0.412 | 1.26 (0.95 to 1.67) | 0.114 |
| Ethnicity (Montubio or other) | 1.01 (0.83 to 1.24) | 0.903 | 0.82 (0.76 to 0.89) | <0.001 | 0.97 (0.83 to 1.14) | 0.718 |
| Basic education of the children (no formal education is the ref.) | 2.52 (0.69 to 9.18) | 0.160 | 0.82 (0.50 to 1.35) | 0.433 | | |
| Economic quintiles by income[a] | | | | | | |
| Economic quintile by income (1st quintile is the ref.) | 1 | - | 1 | - | 1 | - |
| Economic quintile by income (2nd quintile) | 1.21 (1.14 to 1.29) | <0.001 | 1.13 (1.07 to 1.19) | <0.001 | 1.17 (1.12 to 1.21) | <0.001 |
| Economic quintile by income (3rd quintile) | 1.47 (1.24 to 1.75) | <0.001 | 1.32 (1.05 to 1.64) | 0.015 | 1.33 (1.11 to 1.59) | 0.002 |
| Economic quintile by income (4th quintile) | 1.59 (1.36 to 1.86) | <0.001 | 1.34 (1.10 to 1.63) | 0.004 | 1.39 (1.18 to 1.65) | <0.001 |
| Economic quintile by income (5th quintile) | 1.67 (1.50 to 1.85) | <0.001 | 1.41 (1.19 to 1.67) | <0.001 | 1.39 (1.29 to 1.51) | <0.001 |
| p-for-trend | 1.15 (1.10 to 1.19) | <0.001 | 1.09 (1.05 to 1.14) | <0.001 | 1.09 (1.05 to 1.14) | <0.001 |
| Regular class attendance (otherwise is the ref) | 1.48 (0.94 to 2.34) | 0.091 | 0.67 (0.26 to 1.71) | 0.403 | - | - |
| The head of the household receives the BDH (otherwise is the ref.) | 0.77 (0.74 to 0.81) | <0.001 | 0.97 (0.86 to 1.08) | 0.563 | - | - |
| Number of people in the household (per each extra person) | 0.92 (0.91 to 0.93) | <0.001 | 0.94 (0.91 to 0.97) | <0.001 | 0.93 (0.91 to 0.95) | <0.001 |
| Inadequate disposal of excreta (otherwise is the ref.) | 0.73 (0.68 to 0.78) | <0.001 | 0.87 (0.75 to 1.00) | 0.058 | 0.82 (0.76 to 0.90) | <0.001 |
| Regular physical activity (otherwise is the ref.) | 0.76 (0.75 to 0.78) | <0.001 | 0.77 (0.76 to 0.78) | <0.001 | 0.79 (0.75 to 0.82) | <0.001 |
| Perception of low consumption of vegetables (otherwise is the ref.) | 0.91 (0.79 to 1.05) | 0.196 | 1.01 (0.91 to 1.13) | 0.824 | - | - |
| Days per week of consumption in fast food restaurants (per each extra day of consumption) | 1.05 (1.03 to 1.07) | <0.001 | 1.01 (0.96 to 1.06) | 0.810 | - | - |
| Days per week of school food consumption (per each extra day of consumption) | 0.98 (0.94 to 1.02) | 0.318 | 0.99 (0.93 to 1.06) | 0.751 | - | - |
| The child buys food at school (otherwise is the ref.) | 1.17 (0.95 to 1.44) | 0.139 | 1.10 (0.90 to 1.36) | 0.349 | - | - |
| Consumption of food provided by the school (otherwise is the ref.) | 0.83 (0.73 to 0.95) | 0.007 | 0.82 (0.74 to 0.90) | <0.001 | 0.93 (0.82 to 1.06) | 0.288 |
| Family members recognize, understand, and use the labelling of processed foods (otherwise is the ref.) | 1.24 (1.14 to 1.35) | <0.001 | 1.14 (1.09 to 1.19) | <0.001 | 1.14 (1.02 to 1.26) | 0.019 |

*(Continued)*

**Table 2.** (Continued)

| Variable | Crude models | | Adjusted models | | | |
|---|---|---|---|---|---|---|
| | OR (IC95%) | p-value | Saturated | p-value | Parsimonious | p-value |
| | | | aOR (IC95%) | | aOR (IC95%) | |
| In the family, they reduced the consumption of processed foods with a red label (otherwise is the ref.) | 1.14 (1.10 to 1.18) | <0.001 | 1.08 (0.94 to 1.25) | 0.260 | - | - |

BDH = Human Development Voucher, by its Spanish spelling

[a] Income quintiles are calculated at the household level using monetary labour income per capita, first calculating the total income for each income earner. This total income includes earnings from work, income from investments, transfers, and other benefits, such as cash social transfers. Once we add all these up, we obtain the total household income. Then, we determine the average income per person (per capita income) by dividing the total household income by the number of people in each household. Subsequently, the population is systematically arranged on the basis of the per capita income variable. The calculation of the quintiles was performed by dividing the population into five equal groups, known as quintiles. The first quintile includes the percentage of households with the lowest income, the second quintile includes the next percentage, and so on until the fifth quintile, which includes the percentage of households with the highest income.

(aOR) of 1.14 (95% CI: 1.02 to 1.26). Conversely, the consumption of food provided by schools was linked with a non-significant reduction in the risk of overweight or obesity, with an aOR of 0.93 (95% CI: 0.82 to 1.06).

In analysing the determinants of overweight and obesity in children, notable differences emerged between genders when running the final parsimonious model (Table 3). For each incremental year in age, a significant association with overweight or obesity was noted in both genders, with the odds of being overweight or obese increasing by an adjusted odds ratio (aOR) of 1.09 (95% CI: 1.07 to 1.11) for women and 1.10 (95% CI: 1.09 to 1.11) for men. Among the ethnicities, White ethnicity was associated with the highest risk in women, with an aOR of 1.57 (95% CI: 1.34 to 1.83). With respect to economic quintiles, women in the 5th quintile exhibited the greatest risk, with aOR of 1.38 (95% CI: 1.13 to 1.70). In relation to other determining factors, it was observed that inadequate disposal of excreta had significant associations with overweight or obesity in both genders. For women, the risk was reduced with an adjusted odds ratio (aOR) of 0.86 (95% CI: 0.80 to 0.92), whereas for men, the reduction in risk was slightly more pronounced with an aOR of 0.81 (95% CI: 0.74 to 0.89). Regular physical activity appeared to be protective against overweight or obesity. Women who engaged in regular physical activity presented a reduced risk, as indicated by an aOR of 0.84 (95% CI: 0.80 to 0.88), whereas men benefitted slightly more from such activity, displaying an aOR of 0.76 (95% CI: 0.70 to 0.83). Notably, in households where processed food labelling was recognized, understood, and utilized, the risk of overweight or obesity increased in both women and men. This association, for women, was statistically significant with an aOR of 1.16 (95% CI: 1.09 to 1.24), and for men, it was not significant (aOR = 1.11; 95% CI: 0.95 to 1.29).

After conducting an analysis excluding children in the highest quintile, children with parents receiving the BDH, and those residing in households with a larger number of individuals, the observed associations maintained similar trends. However, the relationships were less statistically significant, as detailed in S3 Table.

## Discussion

### Principle findings

Increasing age, male gender, mestizo (mixed ethnic background) ethnicity, higher economic quintiles, inadequate disposal of excreta, and lack of physical activity are factors associated with a higher likelihood of overweight or obesity in children aged 5 to 11 years in Ecuador. The impact of consuming school-provided meals was inconclusive. Children from families

**Table 3. Adjusted Odds Ratios of overweight or obesity from each explanatory variable using the parsimonious logistic regression model of Table 2 between women and men.**

| Variable | Parsimonious model in women | p-value | Parsimonious model in men | p-value |
|---|---|---|---|---|
| | aOR (IC95%) | | aOR (IC95%) | |
| Age of the child (per each increase in one year) | 1.09 (1.07 to 1.11) | <0.001 | 1.10 (1.09 to 1.11) | <0.001 |
| Ethnicity | | | | |
| Ethnicity (Indigenous is the ref.) | 1 | - | 1 | - |
| Ethnicity (Afroecuadorian) | 1.19 (0.82 to 1.72) | 0.374 | 1.07 (0.93 to 1.23) | 0.325 |
| Ethnicity (Mestizo–mixed ethnic background) | 1.32 (1.09 to 1.60) | 0.005 | 1.01 (0.97 to 1.06) | 0.653 |
| Ethnicity (White) | 1.57 (1.34 to 1.83) | <0.001 | 1.05 (0.64 to 1.70) | 0.858 |
| Ethnicity (Montubio or other) | 0.94 (0.67 to 1.32) | 0.716 | 0.99 (0.97 to 1.00) | 0.196 |
| Economic quintiles by income[a] | | | | |
| Economic quintile by income (1st quintile is the ref.) | 1 | - | 1 | - |
| Economic quintile by income (2nd quintile) | 1.13 (1.10 to 1.17) | <0.001 | 1.19 (1.14 to 1.24) | <0.001 |
| Economic quintile by income (3rd quintile) | 1.27 (0.88 to 1.83) | 0.203 | 1.38 (1.32 to 1.44) | <0.001 |
| Economic quintile by income (4th quintile) | 1.31 (1.07 to 1.60) | 0.009 | 1.48 (1.26 to 1.73) | <0.001 |
| Economic quintile by income (5th quintile) | 1.38 (1.13 to 1.70) | 0.002 | 1.40 (1.33 to 1.48) | <0.001 |
| p-for-trend | 1.09 (1.01 to 1.17) | 0.021 | 1.10 (1.09 to 1.12) | <0.001 |
| Number of people in the household (per each extra person) | 0.92 (0.90 to 0.95) | <0.001 | 0.93 (0.88 to 1.00) | 0.041 |
| Inadequate disposal of excreta (otherwise is the ref.) | 0.86 (0.80 to 0.92) | <0.001 | 0.81 (0.74 to 0.89) | <0.001 |
| Regular physical activity (otherwise is the ref.) | 0.84 (0.80 to 0.88) | <0.001 | 0.76 (0.69 to 0.83) | <0.001 |
| Consumption of food provided by the school (otherwise is the ref.) | 1.01 (0.98 to 1.05) | 0.499 | 0.86 (0.69 to 1.09) | <0.001 |
| Family members recognize, understand, and use the labelling of processed foods (otherwise is the ref.) | 1.16 (1.09 to 1.24) | <0.001 | 1.11 (0.95 to 1.29) | 0.179 |

BDH = Human Development Voucher, by its Spanish spelling

[a] Income quintiles are calculated at the household level using monetary labour income per capita, first calculating the total income for each income earner. This total income includes earnings from work, income from investments, transfers, and other benefits, such as cash social transfers. Once we add all these up, we obtain the total household income. Then, we determine the average income per person (per capita income) by dividing the total household income by the number of people in each household. Subsequently, the population is systematically arranged on the basis of the per capita income variable. The calculation of the quintiles was performed by dividing the population into five equal groups, known as quintiles. The first quintile includes the percentage of households with the lowest income, the second quintile includes the next percentage, and so on until the fifth quintile, which includes the percentage of households with the highest income.

with a higher number of individuals in the household and with families that recognize and use processed food labels exhibited a higher likelihood of being overweight or obese.

## Comparison with the literature

The prevalence of OW/OB in school children places the country in the eighth position in the Americas, following countries such as Mexico, Chile, Panama, and the United States, and surpassing the prevalences in Colombia and Peru [23]. International data reported in 2016 ranked the country 15th in the same region [24]. This highlights the drastic increase in prevalence in the absence of effective public policies, compared with other countries that have taken strong measures against childhood overweight and obesity [25, 26].

The study reveals that children who purchase food at school are at a greater risk of being overweight or obese than those who consume meals provided by the school, but this difference is not statistically significant. Despite that, it is important to mention that prior research from the ENSANUT 2018 survey, indicated a link between school foods, particularly those sold in stores (73%), and elevated BMI [27]. A plausible explanation is that 60% of these stores provide unhealthy products labelled with a "red traffic light". The lack of significant results may stem

from the Ecuadorian school feeding program, which relies on processed and packaged products mandated to bear nutritional labelling (traffic light system) [27]. These products contain high levels of sugar, salt, and fats similar to the ultra-processed items distributed by food industries in formal markets [27].

The study underscores the crucial role of the school food environment in addressing childhood obesity. Despite robust evidence [28], many Latin American schools, including those in Ecuador [29], lack effective public health policies. Practical recommendations include prohibiting the advertisement and sale of processed foods in and around schools, imposing higher taxes on sugar-rich, ultra-processed products, and promoting the consumption of fresh fruits and vegetables, along with education on food labelling [30].

In a broader epidemiological context, the gradual increase in the prevalence of OW/OB in the population of this study, along with a population of adults who have been overweight and obese for decades, coupled with a distribution of excess malnutrition across all socioeconomic strata in both children and adults [7], places Ecuador in a second stage of transitioning to obesity [6]. It is noteworthy that the richest quintile of households has a lower prevalence of OW/OB than the quintile immediately below, which could indicate that Ecuador is entering the third stage of transitioning to obesity, where excess malnutrition decreases in wealthier households and becomes concentrated in socioeconomically vulnerable households [6]. Additionally, the observation that children residing in homes with inadequate excreta disposal are less likely to be OW/OB underscores the presence of a socioeconomic gradient in obesity and overweight prevalence in Ecuador. This characteristic is compounded by the fact that the country has not overcome malnutrition in early childhood [7], further complicating the issue of overall malnutrition, resulting in a country experiencing a double burden of malnutrition (malnutrition due to deficiency and excess).

The prevalence patterns of OW/OB in Ecuadorian school children are similar to those in other low and middle-income countries in the region. Similar to a study of school children in Mexico, there is a higher prevalence of OW/OB in boys than in girls, in the non-indigenous population, and in higher-income households, and OW/OB also increases with age [31, 32]. However, the results of this study differ from patterns seen in high-income countries in the region such as Chile, where OW/OB is more prevalent in socioeconomically vulnerable families [33]. It is challenging to compare the determinants found in this study with those in other countries in the region because of to the scarcity of published studies [34].

The fact that children from households self-identified as mestizo have higher rates of OW/OB may be explained by the fact that these children tend to live in urban environments where they are more exposed to obesogenic environments such as advertisements for sugary beverages, ultra-processed foods, and junk food. This would also explain why indigenous children are protected from obesity and overweight, in addition to socioeconomic vulnerabilities in this population, along with household size, which is more strongly linked to malnutrition due to deficiency rather than excess in the country [13, 35]. Therefore, further research is needed on the circumstances of social vulnerability and food insecurity in households of school children with OW/OB to explain why certain negative socioeconomic factors seem to protect against OW/OB.

Physical activity among schoolchildren acts as a protective factor against OW/OB [36, 37]. However, the percentage of children in the sample who engage in regular physical activity is very low, which aligns with previously published findings [38]. There is limited research linking physical activity to excess malnutrition in this group. Therefore, further research on this topic is necessary [39].

Regarding gender differences, our findings support the current understanding of gender-based dietary segregation, both in Ecuador and globally. In Ecuador, prevailing dietary

customs, shaped by a patriarchal and sexist societal framework, often prioritize feeding boys. This may partly account for why physical activity appears to benefit boys more significantly in reducing the likelihood of being overweight or obese, as compared to girls, where such activities do not seem to diminish the probability of overweight or obesity as effectively [40]. This is exacerbated by the inclination to maintain purely aesthetic standards, which results in food restrictions for girls, especially as they enter adolescence [41]. Additionally, there are existing and differential unhealthy exposures based on socioeconomic strata [41].

Our analysis uncovers a 14% increase in obesity and overweight risk among individuals interacting with the nutritional traffic light labelling system, casting light on its ineffectiveness in reducing the intake of processed and ultra-processed foods. This finding not only challenges the system's foundational goal but also implies a significant, albeit counterintuitive, relationship between frequent exposure to the labelling system and an elevated prevalence of obesity and overweight. Such an association suggests that engagement with the system, rather than mitigating risks, may inadvertently contribute to higher obesity rates. Additionally, our research highlights a stark gender disparity in the impact of the labelling system, with women experiencing a markedly higher risk of obesity and overweight than men. This gender-based difference underscores the unique vulnerability of women, potentially linked to either limited [42] or less intense physical activity opportunities [43], thus exacerbating their risk in the face of processed food consumption.

Given Ecuador's extensive history of childhood chronic malnutrition, a pivotal risk factor for overweight, obesity, and non-communicable diseases, coupled with the severe impacts of climate change linked to environmental factors, it is imperative to focus research and public policies on understanding the global syndemic of malnutrition. This approach should incorporate triple-duty actions, considering the distinctive aspects of diverse life stages [44].

Our findings suggest that public health policies should place greater focus on improving the quality of available foods within schools to mitigate the risks of childhood overweight and obesity.

In the context of Ecuador and the wider region, our study underscores the criticality of scrutinizing policy missteps that deregulate the marketing of unhealthy foods and other harmful products, thereby hindering the enhancement of food environments. Remarkably, the President of Ecuador enacted Presidential Decree No. 645 on January 10, 2023, which aims to slash taxes on known health hazards, such as alcohol, tobacco, sugary beverages, and firearms, in stark contrast to advancing public health policies [14]. This decree is poised to adversely affect collective health. This underscores a significant lacuna in public health strategies, potentially exacerbating the obesity and overweight crisis among school children. This situation urgently demands the attention of authorities and policymakers to realign regulations with public health principles, ensuring that economic interests and conflicts of interest do not undermine the development of sound public health policies [45, 46]. In this context, it becomes imperative for Ecuadorian authorities to implement and strengthen comprehensive policies that address both the quality of food offered within schools and those sold in their vicinity.

The congruence between the primary and sensitivity analyses bolsters the robustness of our findings. Although the consumption of food purchased at school and food provided by the school did not attain statistical significance in the model adjusted for expansion factors, the overall pattern of results remained consistent. Importantly, the existing scientific literature supports our findings concerning these variables. It is also crucial to emphasize that there are additional food environments associated with formal markets and marketing targeted at children, which were not considered in this study but significantly contribute to childhood OW/OB. Consequently, it is imperative that national health and nutrition surveys not only possess sensitivity to growth retardation but also to excess malnutrition. While the study delves into

various sociodemographic factors, some potentially relevant variables, such as cultural practices or parental education, are not thoroughly examined. It is worth mentioning that not all information from different forms can be cross-referenced for analysis because of variations in the design and objectives of the national survey.

We believe that our findings necessitate authorities to contemplate public policies aimed at reducing the burden of OW/OB from an early age, thereby diminishing future disabilities and deaths, which evidently result in heavier economic costs and social burdens.

The study calls for comprehensive policy actions to curb the marketing and availability of unhealthy foods, including strict regulation of advertising and sales practices for sugary beverages and other unhealthy products. It critiques policy missteps like Presidential Decree 645, which reduce taxes on health hazards, potentially exacerbating the obesity crisis by making such products more affordable. Highlighting the need for regulations to adhere to public health principles, the research advocates for a multifaceted strategy to improve food quality in and around schools, emphasizing the importance of evidence-based policymaking and prioritizing public health to effectively tackle childhood overweight and obesity in Ecuador.

## Strengths and limitations

The foremost strength of our study lies in its utilization of the nationally representative ENSANUT 2018 survey, which significantly enhances the credibility of our analysis and offers a genuine reflection of the child population in Ecuador. By adopting WHO criteria for evaluating overweight and obesity, we not only ensure the accuracy of our measurements but also improve the comparability of our findings regarding their prevalence and determinants with those of other nations and contexts. Additionally, our use of multilevel modelling techniques in the statistical analysis deepens our insight, adeptly addressing the complexity of the data structure, such as the distribution of children across various regions. This approach allows for a nuanced exploration of the collective impact of diverse factors on the rates of overweight and obesity. Moreover, the extensive scope of the ENSANUT 2018 survey, encompassing a broad array of demographic, socioeconomic, and health-related factors reflective of conditions common to numerous countries beyond Ecuador, supports the external validity of our study. This suggests that our findings have the potential to be applicable in other nations with analogous contexts. Consequently, our research provides critical epidemiological insights into childhood overweight and obesity in Ecuador, while also serving as an invaluable tool for guiding public health strategies in other countries confronting similar health issues.

Given the reliance of this study on secondary analysis of existing survey data, it is crucial to acknowledge the limitations posed by the variables available from the survey, which may not capture the full spectrum of contextual influences on childhood obesity. These include cultural, environmental, and policy-related factors that significantly affect obesity prevalence among children. Additionally, the potential exists for other important, yet unmeasured or unconsidered, factors to contribute, such as genetic predispositions, past medical conditions (e.g., hypothyroidism), medication use, detailed eating behaviours, parental practices, and community-level interventions. Moreover, while our study establishes correlations between certain demographic and socioeconomic factors and childhood obesity, the observational cross-sectional design precludes the determination of causality.

## Conclusions

Increasing age, male gender, mestizo (mixed ethnic background) ethnicity, higher economic quintiles, inadequate disposal of excreta, and lack of physical activity are factors associated with a higher likelihood of overweight or obesity in children aged 5 to 11 years in Ecuador.

The impact of consuming school-provided meals was inconclusive. Children from families with that recognize and use processed food labels exhibited a higher likelihood of being over-weight or obese; this indicates that the nutritional traffic light labelling system, contrary to its intended purpose, is linked to a 14% heightened risk of obesity and overweight, especially among women, highlighting its limited efficacy and underscoring the urgency to reinforce this public health strategy. Our findings underscore the need for a critical reassessment of Ecua-dor's public health policies, emphasizing the improvement of school food quality and stricter regulation of unhealthy product marketing to mitigate childhood overweight and obesity risks through school-based dietary interventions.

## Supporting information

**S1 Data.**
(CSV)

**S1 Table. STROBE statement—Reporting checklist for cross sectional study.**
(DOCX)

**S2 Table. Characteristics comparison between non-overweight/non-obese and overweight/ obese children.** Each category's percentages highlight the proportional differences between the two groups.
(DOCX)

**S3 Table. Adjusted Odds Ratios of overweight or obesity from each explanatory variable using the parsimonious logistic regression model of Table 2 after excluding: (i) people categorized in the lowest income quintile, (ii) people categorized in the highest income quintile, (iii) people who receive the BDH, (iv) people within the upper third of the highest number of people per household.**
(DOCX)

## Acknowledgments

We extend our sincere gratitude to David Grijalva, an economics student from the Faculty of Economics at the Pontifical Catholic University of Ecuador, for his invaluable assistance in the preparation of the tables for this manuscript. Additionally, we would like to thank Natali Men-doza and Margoth Herrera from the Department of Sociodemographic Statistics at the National Institute of Statistics and Censuses for their specific technical guidance on the online database.

## Author Contributions

**Conceptualization:** Betzabé Tello, José Ocaña, Paúl García-Zambrano, Betsabé Enríque-Mor-eira, Iván Dueñas-Espín.

**Data curation:** Betzabé Tello, José Ocaña, Iván Dueñas-Espín.

**Formal analysis:** Betzabé Tello, Iván Dueñas-Espín.

**Funding acquisition:** Betzabé Tello.

**Investigation:** Betzabé Tello, Iván Dueñas-Espín.

**Methodology:** Betzabé Tello, José Ocaña, Iván Dueñas-Espín.

**Project administration:** Betzabé Tello.

**Resources:** Betzabé Tello, José Ocaña, Iván Dueñas-Espín.

**Software:** Betzabé Tello, José Ocaña, Iván Dueñas-Espín.

**Supervision:** Betzabé Tello, Iván Dueñas-Espín.

**Validation:** Betzabé Tello, José Ocaña, Paúl García-Zambrano, Betsabé Enríque-Moreira, Iván Dueñas-Espín.

**Visualization:** Betzabé Tello, Iván Dueñas-Espín.

**Writing – original draft:** Betzabé Tello, José Ocaña, Iván Dueñas-Espín.

**Writing – review & editing:** Betzabé Tello, José Ocaña, Paúl García-Zambrano, Betsabé Enríque-Moreira, Iván Dueñas-Espín.

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
