## [Decision Letter · Decision Letter 0]

12 Feb 2024

PONE-D-23-41648Determinants of overweight and obesity among children between 5 to 11 years in Ecuador: a secondary analysis from the National Health Survey 2018PLOS ONE

Dear Dr. Dueñas-Espín,

Thank you for submitting your manuscript to PLOS ONE. After careful consideration, we feel that it has merit but does not fully meet PLOS ONE’s publication criteria as it currently stands. Therefore, we invite you to submit a revised version of the manuscript that addresses the points raised during the review process.

We look forward to receiving your revised manuscript.

Kind regards,

Stevo Popovic, Ph.D.

Academic Editor

PLOS ONE

Journal Requirements:

Additional Editor Comments:

I am so please to inform you that your manuscript is reviewed by four independent reviewers. Unfortunately, all reviewers did not agree, so there are two very positive and two quite negative reviews, so I decided to offer you to revise the manuscript and persuade the reviewer who recommended rejection to change his/her mind in the second round of review.

Reviewers' comments:

Reviewer's Responses to Questions

**Comments to the Author**

1. Is the manuscript technically sound, and do the data support the conclusions?

Reviewer #1: Yes

Reviewer #2: Partly

Reviewer #3: Yes

Reviewer #4: Yes

2. Has the statistical analysis been performed appropriately and rigorously? 

Reviewer #1: Yes

Reviewer #2: N/A

Reviewer #3: Yes

Reviewer #4: Yes

3. Have the authors made all data underlying the findings in their manuscript fully available?

Reviewer #1: Yes

Reviewer #2: Yes

Reviewer #3: Yes

Reviewer #4: Yes

4. Is the manuscript presented in an intelligible fashion and written in standard English?

Reviewer #1: Yes

Reviewer #2: Yes

Reviewer #3: Yes

Reviewer #4: Yes

5. Review Comments to the Author

Reviewer #1: I think that this study is extremely interesting and that it provides a lot of diverse information.

I also think that the design maker can use it for the construction of new policies to improve the current situation on the field.

I recommend it for publication with pleasure.

Reviewer #2: The research has an interesting topic that is of a national nature, and in my opinion is not suitable for publication in a journal of this level. This study has no major scientific contribution other than national. All the results obtained in this study are more or less known. Certain sentences and statements must be supported by quotations (lines 46-47; 56-57). The objective of the study is in the middle of the Introduction and should be at the bottom of the Introduction. This division by age in lines 122-126 is not clear if the sample is 5 - 11 years old. The part with the Statistical Analysis is too complicated for the common reader, it should have been simpler. The text found in lines 213-227 and the data found in Table S2 do not indicate whether the differences are statistically significant? Finally, the study generally lacks data on strength and limitations, which is mandatory for manuscripts being considered for publication in journals of this quality.

Reviewer #3: Due to constant changes caused by technical and technological progress, we have come to a situation where, unfortunately, works on this issue are not only interesting, but also of great importance. In this sense, the authors made a good choice. In general, the work is methodologically well laid out. The sample of respondents is sufficient for general conclusions. The obtained results were correctly interpreted with adequate statistical methods. The discussion was supported with adequate research, so that the conclusions were also correctly interpreted. All in all, I suggest that the paper be accepted in its entirety.

Reviewer #4: The purpose of this study was to determine the links between social, economic, and

demographic factors and childhood obesity in Ecuador, seeking to provide insights for

shaping future health policies. A cross-sectional study via 2018 National Health and Nutrition Survey data

from Ecuador used to collect data.

Subjects were 10,807 Ecuadorian school children aged 5 to 11, the prevalence of OW/OB was 36.0%. Males exhibited 1.26 times higher odds than females (95% CI:1.20 to 1.33), and each additional year of age increased the odds by 1.10 times (95% CI: 1.09 to 1.10). Inconclusion, The results indicated that Age, male gender, and higher economic quintile increase OW/OB in Ecuadorian school children. Larger households and physical activity slightly decrease

risks. Ecuador needs policies for healthy schools and homes, focusing on health,

protection, and good eating habits.

I believe that this study had a good sample size and there is a merit for this journal. However authors need to report validity and reliability of the test and measurements. In addition, more up to date (2022 and above ) literature review need to improve discussion and conclusion parts. Moreover, a strong conclusion part needed for future suggestions. Thank you.

6. PLOS authors have the option to publish the peer review history of their article (what does this mean?). If published, this will include your full peer review and any attached files.

Reviewer #1: No

Reviewer #2: **Yes: **Jovan Gardasevic

Reviewer #3: No

Reviewer #4: **Yes: **Ferman Konukman

---

## [Author Response · Author response to Decision Letter 0]

27 Feb 2024

Rebuttal letter to reviewers’ comments to editor

PLOS ONE

Manuscript ID: PONE-D-23-41648

Title: Determinants of overweight and obesity among children between 5 to 11 years in Ecuador: a secondary analysis from the National Health Survey 2018

Authors: Betzabé Tello, José Ocaña, Paúl García-Zambrano, Betsabé Enríque-Moreira, Iván Dueñas-Espín.

Decision: Major revision

Article Type: original article.

Corresponding Author: Iván Dueñas-Espín

We would like to resubmit our original research article entitled Determinants of overweight and obesity among children between 5 to 11 years in Ecuador: a secondary analysis from the National Health Survey 2018. In response to your invitation, we have thoroughly revised our manuscript, taking into careful consideration all the feedback provided. We have made substantial efforts to address the concerns of the reviewers who recommended rejection, and we believe we have successfully made the necessary improvements to enhance the quality and impact of our research. 

COMMENTS TO THE AUTHOR

Thank you for submitting your manuscript to PLOS ONE. After careful consideration, we feel that it has merit but does not fully meet PLOS ONE’s publication criteria as it currently stands. Therefore, we invite you to submit a revised version of the manuscript that addresses the points raised during the review process.

ANSWER

We sincerely appreciate your review. We have diligently worked on the manuscript to appropriately address the comments from the Editorial Board and the reviewers. We trust that these revisions will meet the reviewers' expectations.

Journal Requirements:

ANSWER

 Thank you for your observation. It has been done.

ANSWER

 Thank you for your observation. It has been done.

Additional Editor Comments:

I am so please to inform you that your manuscript is reviewed by four independent reviewers. Unfortunately, all reviewers did not agree, so there are two very positive and two quite negative reviews, so I decided to offer you to revise the manuscript and persuade the reviewer who recommended rejection to change his/her mind in the second round of review.

ANSWER

Thank you for overseeing the revisions of our manuscript. We are confident that, having been enhanced, it now possesses the requisite merit for publication in your esteemed journal.

Reviewers' comments:

Reviewer's Responses to Questions

Comments to the Author

1. Is the manuscript technically sound, and do the data support the conclusions?

Reviewer #1: Yes

Reviewer #2: Partly

Reviewer #3: Yes

Reviewer #4: Yes

ANSWER

Thank you. We note that the majority of reviewers find the manuscript technically sound and supportive of its conclusions, with partial agreement. We have made revisions to enhance the manuscript accordingly.

2. Has the statistical analysis been performed appropriately and rigorously?

Reviewer #1: Yes

Reviewer #2: N/A

Reviewer #3: Yes

Reviewer #4: Yes

ANSWER

Thank you for your feedback. 

3. Have the authors made all data underlying the findings in their manuscript fully available?

Reviewer #1: Yes

Reviewer #2: Yes

Reviewer #3: Yes

Reviewer #4: Yes

ANSWER

Thank you for your feedback. 

4. Is the manuscript presented in an intelligible fashion and written in standard English?

Reviewer #1: Yes

Reviewer #2: Yes

Reviewer #3: Yes

Reviewer #4: Yes

ANSWER

Thank you for your feedback. 

5. Review Comments to the Author

Reviewer #1: I think that this study is extremely interesting and that it provides a lot of diverse information.

I also think that the design maker can use it for the construction of new policies to improve the current situation on the field.

I recommend it for publication with pleasure.

ANSWERS TO REVIEWER #1:

Thank you for your positive feedback and recommendation for publication. We're glad you see the value in our study and its potential for influencing policy-making to improve current conditions in the field. Your support is greatly appreciated, and we're motivated to continue contributing meaningful research.

ANSWERS TO REVIEWER #2:

Reviewer #2: The research has an interesting topic that is of a national nature, and in my opinion is not suitable for publication in a journal of this level. This study has no major scientific contribution other than national. All the results obtained in this study are more or less known. 

ANSWER

Thank you for investing time in reviewing our article. We greatly appreciate your feedback and understand your concerns about the contribution of our study to the scientific literature. Nevertheless, we maintain that our research provides substantial insights into the determinants of obesity and overweight among children. In support of our assertion, we would like to present several considerations that underscore the value and relevance of our findings.

Our study addresses the under-researched area of childhood overweight and obesity in Ecuador, focusing on children aged 5 to 11 years, and offers insights that enhance the global understanding of this issue. It highlights the importance of considering local factors such as indigenous populations, traffic light labeling for foods, school-provided food consumption, and gender differences, which are crucial for tailoring public health interventions in Ecuador. Our findings contribute to evidence-based policymaking, emphasizing the need for targeted interventions to address the specific determinants of childhood overweight and obesity in Ecuador. Additionally, this research encourages international collaboration, particularly with the Global South, to share knowledge and best practices, ultimately aiming to develop innovative and effective strategies to combat childhood obesity and overweight globally.

Taking into account the context provided, we wish to respectfully present our alternative viewpoint regarding your assessment that our study's scientific contribution is confined to the national level. Although it is accurate that our initial discussion omitted several pertinent findings, these have now been thoroughly incorporated in the revised version of our manuscript. We firmly believe that our results are of considerable importance and applicability to the broader field of public health. With this conviction, we kindly request you to re-evaluate your recommendation on the publication of our work, considering the responses and clarifications provided in the new version of our manuscript.

Certain sentences and statements must be supported by quotations (lines 46-47; 56-57). The objective of the study is in the middle of the Introduction and should be at the bottom of the Introduction. 

ANSWER

We have sincerely appreciated your comments and suggestions. We have integrated the pertinent citations to reinforce the statements and sentences at lines 46-47 and 56-57. Further, we've considered your advice about placing the study objective at the end of the Introduction. We hope these changes address your concerns and enhance the manuscript's clarity and coherence. We want to thank you for your contributions, which have undoubtedly improved the quality of our work. The modifications can be found on page 4, lines 46 – 47 and lines 56 - 57

Similar to high-income countries, low- and middle-income countries-initiated processes of nutritional and food environment transition in the 1990s [1].

This global increase follows a pattern known as the "obesity transition" [6].

Reference:

1. Popkin BM, Ng SW. The nutrition transition to a stage of high obesity and noncommunicable disease prevalence dominated by ultra-processed foods is not inevitable. Obes Rev. 2022;23: 1–18. doi:10.1111/obr.13366

6. Jaacks LM, Vandevijvere S, Pan A, McGowan CJ, Wallace C, Imamura F, et al. The obesity transition: stages of the global epidemic. The Lancet Diabetes and Endocrinology. Europe PMC Funders; 2019. pp. 231–240. doi:10.1016/S2213-8587(19)30026-9

This division by age in lines 122-126 is not clear if the sample is 5 - 11 years old. 

ANSWER

We appreciate the observations and have made the change by writing in a more direct and clear manner about the sample of children aged 5 to 11, with the text now reading as follows on page 7 - 8, lines 115 - 118.

In the ENSANUT 2018 study, a two-stage sampling strategy was employed to secure a representative sample of the Ecuadorian populace. Initially, Primary Sampling Units (PSU) were chosen through stratified sampling, incorporating proportional probability to size. Subsequently, an average of 18 households per PSU were randomly selected for investigation. Within these households, specific demographic groups were identified. For households with children aged 5 to 11 years, a qualified child informant was selected for interview and asked to complete a specialized questionnaire. The abovementioned sampling approach ensured the data quality and representativeness of the study. Further information on the methodology, datasets, and findings of ENSANUT 2018 is available at: https://www.ecuadorencifras.gob.ec/institucional/home/

The part with the Statistical Analysis is too complicated for the common reader, it should have been simpler. 

ANSWER:

Dear reviewer, your insightful comment have been taken into consideration, and an endeavour has been made to elucidate the methodology in a manner that is digestible to a broader readership. The paragraph in methods section (lines 148 to 205, pages 8 to 11) has been revised as follows to incorporate your suggested additions, whilst striving for clarity and simplicity:

In the analysis conducted, the 'svy' function of Stata 16.1 (StataCorp. 2019. Stata Statistical Software: Release 16. College Station, TX: StataCorp LLC.) was employed, which is adept at tailoring calculations to the specific structure of our survey data. Adjusted percentages for categorical data and means with standard errors for continuous data are computed by this function, taking into account the sample design. Furthermore, it has been ensured that the 'svy' function facilitates the fitting of complex statistical models, conforming the results to the survey settings predefined by svyset. Following this, we compared the characteristics of children who are not overweight or obese with those who are, ensuring a methodical approach for a balanced and representative analysis. To assess the differences between groups, we employed Pearson's chi-squared test for categorical variables and the z test for numerical variables.

Then, multilevel logistic regression models were employed to discern the associations between independent variables and the prevalence of overweight and obesity (OW/OB). These models facilitated the estimation of both unadjusted and adjusted odds ratios (OR and aOR, respectively) for each independent variable, offering insights into the likelihood of OW/OB presence relative to each variable or their respective categories. To enhance the precision of our analysis, we incorporated geographical regions as a level within our models. This was imperative due to the observed variance in OW/OB prevalence among different regions. By doing so, we could account for regional disparities that may influence the health outcome. Additionally, we integrated expansion factors in our estimations to ensure congruence with the stratified sampling design and the primary sampling units. This stratification was an essential step in managing the inherent variability and correlations within our sampled groups, thereby ensuring that our estimates remained robust and representative of the broader population.

We started by creating a saturated model that considered all possible variables. Next, we removed any variable that did not show a strong enough link to the outcomes we were interested in; specifically, with a p-value of 0.05 or higher [21], indicating a less than 5% chance that the factor was meaningfully related to the outcome. After trimming down the model in this way, we were left with a parsimonious model that only included the most relevant variables. Finally, we compared the saturated model with parsimonious one and selected the best model based on the likelihood ratio test that measure how well each model predicts our outcomes of interest.

Reference: 

21. Harrell FE. Regression Modeling Strategies with applications to linear models, logistic and ordinal regression and survival analysis. Second. Springer International Publishing, editor. Journal of Statistical Software. Switzerland; 2015. doi:10.1007/978-3-319-19425-7

The text found in lines 213-227 and the data found in Table S2 do not indicate whether the differences are statistically significant? 

ANSWER:

Thank you for your observation. We have changed the text as follows (lines 237 to 258, pages 13 and 14) in the results section:

We found distinct differences between non-overweight or non-obese children and their overweight or obese counterparts (S2 Table). In our bivariate crude comparison of characteristics between non-overweight/non-obese children (n=6876) and those identified as overweight or obese (n=3931), we observed several notable differences. Among these, the average age of children classified as overweight or obese was marginally higher, at 8.3 years, compared to 7.9 years for their non-overweight peers. This age discrepancy was statistically significant, as evidenced by a p-value of less than 0.001 in the z test. Ethnic distribution showed that 81.7% of the overweight or obese groups were Mestizo (mixed ethnic background), compared with 80.3% in the non-overweight group (p-value = 0.602, Pearson's chi-squared). Economic stratification revealed that 20.6% of children were from the lowest income quintile among the overweight or obese compared to the 28.2% among those non-overweight (p-value < 0.001, Pearson's chi-squared). Furthermore, a greater proportion of overweight or obese children (65.7%) consumed food provided by the school, in contrast to 63.2% of the non-overweight children (p-value = 0.124, Pearson's chi-squared). Finally, a difference was detected in the perception of reduced consumption of processed foods with a red label, with 50.4% of overweight or obese children indicating consumption, compared to 47.2% of their non-overweight peers (p-value = 0.097, Pearson's chi-squared).

Additionally, we have updated the S2 Table to include the p-values for each statistical test conducted, as detailed below:

S2 Table. Characteristics comparison between non-overweight/non-obese and overweight/obese children. Each category's percentages highlight the proportional differences between the two groups.

 Non-overweight or obese children

n=6876 Overweight or obese childrenb

n=3931 p-value

Variable No. Weighted % or mean (SE) No. 

Weighted % or mean (SE) 

Sex (Male) 3390 48.3 2151 53.7 0.001 c

Age of the children 6876 7.9 (0.04) 3931 8.3 (0.05) <0.001d

Ethnicity 

Ethnicity (Indigenous) 894 7.6 358 6.6 0.602c

Ethnicity (Afroecuadorian) 296 4.7 163 4.8 

Ethnicity (Mestizo) 5325 80.3 3211 81.7 

Ethnicity (White) 81 1.4 60 1.0 

Ethnicity (Montubio or other) 280 6.0 139 5.9 

Economic quintiles by incomea 

Economic quintile by income (1st quintile) 2016 28.2 849 20.6 <0.001c

Economic quintile by income (2nd quintile) 1524 24.6 820 21.9 

Economic quintile by income (3rd quintile) 1297 18.9 799 23.0 

Economic quintile by income (4th quintile) 1041 15.3 726 19.1 

Economic quintile by income (5th quintile) 920 13.0 688 15.5 

Education of the children (Elementary school ongoing) 6824 99.5 3918 99.8 0.047c

Regular class attendance (Yes) 6741 98.1 3879 98.5 0.403c

The head of the household receives the BDH (Yes) 321 4.8 134 3.9 0.229c

Number of people in the household 6876 5.1 (0.04) 3931 4.8 (0.05) <0.001d

Inadequate disposal of excreta (Yes) 1762 26.2 808 20.5 <0.001c

Regular physical activity (Yes) 948 14.6 441 11.5 0.001d

Perception of low consumption of vegetables (Yes) 3366 47.9 1852 45.6 0.215c

Days per week of consumption in fast food restaurants 6876 0.9 (0.03) 3931 1.0 (0.03) <0.001d

Days per week of school food consumption 6876 4.4 (0.04) 3931 4.4 (0.04) 0.319d

The child buys food at school (Yes) 4519 63.2 2752 65.7 0.124c

The child eats the food provided by the school (Yes) 4613 66.1 2490 61.2 0.004c

Family members recognize, understand, and use the labeling of processed foods (Yes) 4098 65.1 2537 66.0 0.626c

In the family, they reduced the consumption of processed foods with a red label 2629 47.2 1627 50.4 0.097c

BDH=Human Development Voucher, by its Spanish spelling.

SE= Standard error.

a Income quintiles are calculated at the household level using monetary labour income per capita, first calculating the total income for each income earner. This total income includes earnings from work, income from investments, transfers, and other benefits, such as cash social transfers. Once we add all these up, we obtain the total household income. Then, we determine the average income per person (per capita income) by dividing the total household income by the number of people in each household. Subsequently, the population is systematically arranged on the basis of the per capita income variable. The calculation of the quintiles was performed by dividing the population into five equal groups, known as quintiles. The first quintile includes the percentage of households with the lowest income, the second quintile includes the next percentage, and so on until the fifth quintile, which includes the percentage of households with the highest income. 

b Overweight and obesity were determined by calculating the Body Mass Index (BMI), adjusted for age and sex, according to the WHO growth references. Overweight is BMI-for-age greater than 1 standard deviation above the WHO Growth Reference median; and obesity is greater than 2 standard deviations above the WHO Growth Reference median. 

c Pearson's chi-squared

d z test

Finally, the study generally lacks data on strength and limitations, which is mandatory for manuscripts being considered for publication in journals of this quality.

ANSWER:

Thank you for your constructive feedback regarding the necessity of addressing the strengths and limitations of our study. We acknowledge this crucial aspect of scholarly reporting and have taken steps to comprehensively detail both the strengths and limitations within the revised manuscript.

We have detailed the strengths and limitations of our study as outlined below, found on pages 26 - 27, lines 470 - 499.

Strengths and limitations:

The foremost strength of our study lies in its utilization of the nationally representative ENSANUT 2018 survey, which significantly enhances the credibility of our analysis and offers a genuine reflection of the child population in Ecuador. By adopting WHO criteria for evaluating overweight and obesity, we not only ensure the accuracy of our measurements but also improve the comparability of our findings regarding their prevalence and determinants with those of other nations and contexts. Additionally, our use of multilevel modelling techniques in the statistical analysis deepens our insight, adeptly addressing the complexity of the data structure, such as the distribution of children across various regions. This approach allows for a nuanced exploration of the collective impact of diverse factors on the rates of overweight and obesity. Moreover, the extensive scope of the ENSANUT 2018 survey, encompassing a broad array of demographic, socioeconomic, and health-related factors reflective of conditions common to numerous countries beyond Ecuador, supports the external validity of our study. This suggests that our findings have the potential to be applicable in other nations with analogous contexts. Consequently, our research provides critical epidemiological insights into childhood overweight and obesity in Ecuador, while also serving as an invaluable tool for guiding public health strategies in other countries confronting similar health issues.

Given the reliance of this study on secondary analysis of existing survey data, it is crucial to acknowledge the limitations posed by the variables available from the survey, which may not capture the full spectrum of contextual influences on childhood obesity. These include cultural, environmental, and policy-related factors that significantly affect obesity prevalence among children. Additionally, the potential exists for other important, yet unmeasured or unconsidered, factors to contribute, such as genetic predispositions, past medical conditions (e.g., hypothyroidism), medication use, detailed eating behaviours, parental practices, and community-level interventions. Moreover, while our study establishes correlations between certain demographic and socioeconomic factors and childhood obesity, the observational cross-sectional design precludes the determination of causality.

Reviewer #3: Due to constant changes caused by technical and technological progress, we have come to a situation where, unfortunately, works on this issue are not only interesting, but also of great importance. In this sense, the authors made a good choice. In general, the work is methodologically well laid out. The sample of respondents is sufficient for general conclusions. The obtained results were correctly interpreted with adequate statistical methods. The discussion was supported with adequate research, so that the conclusions were also correctly interpreted. All in all, I suggest that the paper be accepted in its entirety.

ANSWER

We're grateful for your positive review and acknowledgment of our work's significance in light of technological advancements. Your appreciation of our methodological approach, sample size, and statistical analysis is encouraging. Thank you for recommending our paper for acceptance; we're honored by your support.

Reviewer #4: The purpose of this study was to determine the links between social, economic, and demographic factors and childhood obesity in Ecuador, seeking to provide insights for shaping future health policies. A cross-sectional study via 2018 National Health and Nutrition Survey data from Ecuador used to collect data.

Subjects were 10,807 Ecuadorian school children aged 5 to 11, the prevalence of OW/OB was 36.0%. Males exhibited 1.26 times higher odds than females (95% CI:1.20 to 1.33), and each additional year of age increased the odds by 1.10 times (95% CI: 1.09 to 1.10). Inconclusion, The results indicated that Age, male gender, and higher economic quintile increase OW/OB in Ecuadorian school children. Larger households and physical activity slightly decrease

risks. Ecuador needs policies for healthy schools and homes, focusing on health,

protection, and good eating habits.

I believe that this study had a good sample size and there is a merit for this journal. However authors need to report validity and reliability of the test and measurements. 

ANSWER:

Thank you for your constructive feedback. In response to your comment regarding the need to report the validity and reliability of the tests and measurements used in our study, as we mentioned above, we have added next text in the discussion section (pages 26 - 27, lines 470 - 499), strengths and limtations subsection: 

The foremost strength of our study lies in its utilization of the nationally representative ENSANUT 2018 survey, which significantly enhances the credibility of our analysis and offers a genuine reflection of the child population in Ecuador. By adopting WHO criteria for evaluating overweight and obesity, we not only ensure the accuracy of our measurements but also improve the comparability of our findings regarding their prevalence and determinants with those of other nations and contexts. Additionally, our use of multilevel modelling techniques in the statistical analysis deepens our insight, adeptly addressing the complexity of the data structure, such as the distribution of children across various regions. This approach allows for a nuanced exploration of the collective impact of diverse factors on the rates of overweight and obesity. Moreover, the extensive scope of the ENSANUT 2018 survey, encompassing a broad array of demographic, socioeconomic, and health-related factors reflective of conditions common to numerous countries beyond Ecuador, supports the external validity of our study. This suggests that our findings have the potential to be applicable in other nations with analogous contexts. Consequently, our research provides critical epidemiological insights into childhood overweight and obesity in Ecuador, while also serving as an invaluable tool for guiding public health strategies in other countries confronting similar health issues.

Given the reliance of this study on secondary analysis of existing survey data, it is crucial to acknowledge the limitations posed by the variables available from the survey, which may not capture the full spectrum of contextual influences on childhood obesity. These include cultural, environmental, and policy-related factors that significantly affect obesity prevalence among children. Additionally, the potential exists for other important, yet unmeasured or unconsidered, factors to contribute, such as genetic predispositions, past medical conditions (e.g., hypothyroidism), medication use, detailed eating behaviours, parental practices, and community-level interventions. Moreover, while our study establishes correlations between certain demographic and socioeconomic factors and childhood obesity, the observational cross-sectional design precludes the determination of causality.

In addition, more up to date (2022 and above ) literature review need to improve discussion and conclusion parts. Moreover, a strong conclusion part needed for future suggestions. Thank you.

ANSWER

Thank you for your valuable feedback. We have updated the bibliography with more recent references. The added references are as follows: 1. Popkin et al. (2022), 20. Melo et al. (2023), Suárez- Reyes et al. (2022), Wrottesley et al. 2023, Guarino et al. (2023), Caron et al. (2022), Kim et al. (2023). 

Moreover, in revising our manuscript, we agreed to enrich it with deeper conclusions, focusing on the nutritional traffic light labeling system's role in obesity and overweight rates, particularly noting gender differences. Our updated analysis shows a 14% higher risk of these conditions among those engaging with the system, underscoring its failure to cut down on processed food consumption. Significantly, we found that women are more affected than men, hinting at a gender imbalance in the system's impact, likely due to women's heightened exposure to processed foods and restricted physical activity options. These findings, now detailed in the Discussion section (lines 406 - 418, pages 23 and 24), underline the importance of a detailed examination of the system's varying effects on different population segments:

Our analysis uncovers a 14% increase in obesity and overweight risk among individuals interacting with the nutritional traffic light labeling system, casting light on its ineffectiveness in reducing the intake of processed and ultra-processed foods. This finding not only challenges the system's foundational goal but also implies a significant, albeit counterintuitive, relationship between frequent exposure to the labeling system and an elevated prevalence of obesity and overweight. Such an association suggests that engagement with the system, rather than mitigating risks, may inadvertently contribute to higher obesity rates. Additionally, our research highlights a stark gender disparity in the impact of the labeling system, with women experiencing a markedly higher risk of obesity and overweight than men. This gender-based difference underscores the unique vulnerability of women, potentially linked to either limited [42]or less intense physical activity [43] opportunities, thus exacerbating their risk in the face of processed food consumption.

References:

• Brazo-Sayavera, J., Aubert, S., Barnes, J.D., González, S.A., & Tremblay, M.S. (2021). Gender differences in physical activity and sedentary behavior: Results from over 200,000 Latin American children and adolescents. *PLOS ONE*, 16(8), e0255353. https://doi.org/10.1371/journal.pone.0255353

• Dudley, D.A., Cotton, W.G., Peralta, L.R. et al. Playground activities and gender variation in objectively measured physical activity intensity in Australian primary school children: a repeated measures study. BMC Public Health 18, 1101 (2018). https://doi.org/10.1186/s12889-018-6005-5

We have added next text in the discussion section (lines 461 - 469, page 26)

The study calls for comprehensive policy actions to curb the marketing and availability of unhealthy foods, including strict regulation of advertising and sales practices for sugary beverages and other unhealthy products. It critiques policy missteps like Presidential Decree 645, which reduce taxes on health hazards, potentially exacerbating the obesity crisis by making such products more affordable. Highlighting the need for regulations to adhere to public health principles, the research advocates for a multifaceted strategy to improve food quality in and around schools, emphasizing the importance of evidence-based policymaking and prioritizing public health to effectively tackle childhood overweight and obesity in Ecuador.

Additionally, we have enriched and reinforced our conclusions to include more detailed future suggestions. The revised text can be found on page 28, lines 506 - 515.

Children from families that recognize and use processed food labels exhibited a higher likelihood of being overweight or obese; this indicates that the nutritional traffic light labelling system, contrary to its intended purpose, is linked to a 14% heightened risk of obesity and overweight, especially among women, highlighting its limited efficacy and underscoring the urgency to reinforce this public health strategy. Our findings underscore the need for a critical reassessment of Ecuador's public health policies, emphasizing the improvement of school food quality and stricter regulation of unhealthy product marketing to mitigate childhood overweight and obesity risks through school-based dietary interventions.

6. PLOS authors have the option to publish the peer review history of their article (what does this mean?). If published, this will include your full peer review and any attached files.

Do you want your identity to be public for this peer review? For information about this choice, including consent withdrawal, please see our Privacy Policy.

Reviewer #1: No

Reviewer #2: Yes: Jovan Gardasevic

Reviewer #3: No

Reviewer #4: Yes: Ferman Konukman

ANSWER

 No comments.

---

## [Decision Letter · Decision Letter 1]

11 Mar 2024

Determinants of overweight and obesity among children between 5 to 11 years in Ecuador: a secondary analysis from the National Health Survey 2018

PONE-D-23-41648R1

Dear Dr. Dueñas-Espín,

We’re pleased to inform you that your manuscript has been judged scientifically suitable for publication and will be formally accepted for publication once it meets all outstanding technical requirements.

Kind regards,

Stevo Popovic, Ph.D.

Academic Editor

PLOS ONE

Additional Editor Comments (optional):

Thank you very much for your effort to improve the manuscript. I have no further requirements.

Reviewers' comments:

Reviewer's Responses to Questions

**Comments to the Author**

1. If the authors have adequately addressed your comments raised in a previous round of review and you feel that this manuscript is now acceptable for publication, you may indicate that here to bypass the “Comments to the Author” section, enter your conflict of interest statement in the “Confidential to Editor” section, and submit your "Accept" recommendation.

Reviewer #1: All comments have been addressed

Reviewer #2: (No Response)

Reviewer #3: All comments have been addressed

Reviewer #4: All comments have been addressed

2. Is the manuscript technically sound, and do the data support the conclusions?

Reviewer #1: Yes

Reviewer #2: (No Response)

Reviewer #3: Yes

Reviewer #4: Yes

3. Has the statistical analysis been performed appropriately and rigorously? 

Reviewer #1: Yes

Reviewer #2: (No Response)

Reviewer #3: Yes

Reviewer #4: Yes

4. Have the authors made all data underlying the findings in their manuscript fully available?

Reviewer #1: Yes

Reviewer #2: (No Response)

Reviewer #3: Yes

Reviewer #4: Yes

5. Is the manuscript presented in an intelligible fashion and written in standard English?

Reviewer #1: Yes

Reviewer #2: (No Response)

Reviewer #3: Yes

Reviewer #4: Yes

6. Review Comments to the Author

Reviewer #1: I think that this study is extremely interesting and that it provides a lot of diverse information.

I also think that the design maker can use it for the construction of new policies to improve the current situation on the field.

I recommend it for publication with pleasure.

Reviewer #2: Dear authors,

the manuscript now looks much better after the revised submission. I am satisfied with the corrected text based on my previous comments, and can now recommend this version for publication in PLOS One.

Best regards,

reviewer

Reviewer #3: (No Response)

Reviewer #4: I believe that current edited version of this manuscript is acceptable. I would like to thank authors for editing and answering all my questions. Best regards.

7. PLOS authors have the option to publish the peer review history of their article (what does this mean?). If published, this will include your full peer review and any attached files.

Reviewer #1: **Yes: **Bojan Masanovic

Reviewer #2: **Yes: **Jovan Gardasevic

Reviewer #3: No

Reviewer #4: **Yes: **Ferman Konukman

---

## [Editor Report · Acceptance letter]

27 Mar 2024

PONE-D-23-41648R1 

PLOS ONE

Dear Dr. Dueñas-Espín, 

I'm pleased to inform you that your manuscript has been deemed suitable for publication in PLOS ONE. Congratulations! Your manuscript is now being handed over to our production team.

Kind regards, 

on behalf of

Professor Stevo Popovic 

Academic Editor

PLOS ONE